# Methodological and Terminological Issues in Animal-Assisted Interventions: An Umbrella Review of Systematic Reviews

**DOI:** 10.3390/ani10050759

**Published:** 2020-04-27

**Authors:** Antonio Santaniello, Francesca Dicé, Roberta Claudia Carratú, Alessia Amato, Alessandro Fioretti, Lucia Francesca Menna

**Affiliations:** 1Department of Veterinary Medicine and Animal Productions, Federico II University of Naples, via Mezzocannone, 8-80134 Naples, Italy; alessiaamatovet@gmail.com (A.A.); alessandro.fioretti@unina.it (A.F.); luciafrancesca.menna@unina.it (L.F.M.); 2University Centre SinAPSi, Federico II University of Naples, via G.C. Cortese, 29-80133 Naples, Italy; francesca.dice@unina.it (F.D.); carraturobertaclaudia@gmail.com (R.C.C.)

**Keywords:** Animal-assisted therapy, animal-assisted activity, animal-assisted education, dog, Horse, methodology

## Abstract

**Simple Summary:**

Animal-assisted interventions (AAIs) include a wide range of activities aimed at improving the health and well-being of people with the help of pets. Although there have been many studies on the effects of these interventions on animal and human wellbeing and health, univocal data on the methodological aspects, regarding type and duration of intervention, operators, involved animal species, and so on, are still lacking. In this regard, several systematic reviews in the scientific literature have already explored and outlined some methodological aspects of animal-assisted interventions. Therefore, we developed an umbrella review (UR) which summarizes the data of a set of suitable systematic reviews (SRs), in order to clarify how these Interventions are carried out. From our results, it is shown that there is a widespread heterogeneity in the scientific literature concerning the study and implementation of these interventions. These results highlight the need for the development and, consequently, the diffusion of protocols (not only operational, but also research approaches) providing for a univocal use of globally recognized terminologies and facilitating comparison between the numerous experiences carried out and reported in the field.

**Abstract:**

Recently, animal-assisted interventions (AAIs), which are defined as psychological, educational, and rehabilitation support activities, have become widespread in different contexts. For many years, they have been a subject of interest in the international scientific community and are at the center of an important discussion regarding their effectiveness and the most appropriate practices for their realization. We carried out an umbrella review (UR) of systematic reviews (SRs), created for the purpose of exploring the literature and aimed at deepening the terminological and methodological aspects of AAIs. It is created by exploring the online databases PubMed, Google Scholar, and Cochrane Library. The SRs present in the high-impact indexed search engines Web of Sciences and Scopus are selected. After screening, we selected 15 SRs that met the inclusion criteria. All papers complained of the poor quality of AAIs; some considered articles containing interventions that did not always correspond to the terminology they have explored and whose operating practices were not always comparable. This stresses the need for the development and consequent diffusion of not only operational protocols, but also research protocols which provide for the homogeneous use of universally recognized terminologies, thus facilitating the study, deepening, and comparison between the numerous experiences described.

## 1. Introduction

Animal-assisted interventions (AAIs) have been considered by the International Association of Human–Animal Interaction Organizations (IAHAIO) [1] as recreational, educational, or rehabilitation/therapeutic activities which, due to the presence and mediation of domestic animals, aim to act on pathological situations and on social or educational problems. They have been a subject of interest and study in health disciplines for many years [2,3,4,5,6,7], according to the criteria provided by the “One Health–One Medicine Initiatives", promoting collaboration and communication between different disciplines to work together at local, national, and global levels, establishing an integrated approach [8,9,10,11,12,13,14].

More specifically, as reported by the IAHAIO White Paper ([1], p.5), “An animal-assisted intervention is a goal oriented and structured intervention that intentionally includes or incorporates animals in health, education and human services (e.g., social work) for the purpose of therapeutic gains in humans". These interventions incorporate human–animal teams in formal human services and, as such, these interventions should be developed and implemented using an interdisciplinary approach [1]. AAIs include animal-assisted activity (AAA), animal-assisted therapy (AAT), and animal-assisted education (AAE). AAAs are planned and goal-oriented informal interactions and visits conducted by the human–animal team for motivational, educational, and recreational purposes [1]. AAT is defined as a goal-oriented, planned, and structured therapeutic intervention directed and/or delivered by health, education, or human service professionals (e.g., psychologists) and focused on the socio-emotional functioning of the human recipient, either in a group or individual setting. The professional delivering AAT (...) must have adequate knowledge about the behavior, needs, health, and indicators and regulation of stress in the animals involved [1].

AAT can act as a support to psychotherapeutic activities, understood as a collection of rules or techniques used to conduct mental health treatment, having a relevant set of goals between a professional trained person (known as a therapist) and the recipient or subject of the therapy (known as the client or patient) [15]. In the scientific literature, moreover, it has been considered fundamental that the application of this type of intervention, aimed at the treatment of complex psychic conditions, refers to consolidated and structured theoretical reference models, which present precise indications concerning the theory of the technique to be implemented in the examination room; in order to ensure, as far as possible, the replicability of the intervention itself, its success, and the achievement of the proposed objectives [16,17].

AAE is described as goal-oriented, planned, and structured interventions directed and/or delivered by educational (and related) service professionals. AAE is conducted by qualified (i.e., with degree) general and special education teachers, either in a group or individual setting [1]. They act as support for educational interventions, defined in the literature as an action through which individuals develop or perfect intellectual, social, and physical faculties and attitudes [18].

Finally, as reported by the IAHAIO [1], there is also animal-assisted coaching/counseling (AAC), defined as goal-oriented interventions, which are planned, structured, and directed and/or provided by authorized professionals (e.g., coaches or consultants) and assisted by animals. The coach/consultant (...) must have adequate training on the behavior, needs, health, and indicators and stress regulation of the animals involved. They provide support for consultancy activities, interventions aimed at promoting the development and use of the client’s potential, helping them to overcome any personal difficulties in which one person supports another in achieving a specific goal [19].

As above, the various areas in which the AAIs apply (AATs, AAAs, AAE) have been defined with respect to the terminology (although sometimes they often overlap) and there exists various scientific evidence of their effectiveness. Given the complexity and the variety of these interventions, there is still a strong discussion with respect to the definitions (i.e., used terminology) and the corresponding applied methodologies [12,13]. In addition, on one hand, few studies have been carried out with regard to health protocols aimed at guaranteeing the safety of the setting and users/patients involved in the these interventions [12,20,21,22,23,24] and, on the other hand, there are no exact univocal and clear regulations regarding the applied methodologies, the most appropriate practices for their implementation, and the training of AAI operators [2,12]. In the literature, it has often been strongly highlighted that the described methodologies are not always clear and that terminologies are not always univocally used [25]. Furthermore, the presence of poor references to the operating protocols used, variables of interest, effects of the SRs on the scientific community, and results or limits of studies are often lamented [25]. Research designs are often described by anecdotal facts, referring to single cases with few links to theoretical frameworks [26].

This topic also provided motivation for the drafting of numerous systematic reviews (SRs) or Meta-Analyses [25,26]; for this reason, we consider umbrella reviews (URs) of systematic reviews (SRs) [27] to be a fast and effective way of exploring the orientation of the scientific community and getting an idea of the state-of-the-art regarding such a complex topic as this. As a study group aimed at deepening the good practices for these interventions, we have realized this paper with the intention of offering a “snapshot" of this topic to interested readers. For this purpose, we have explored the literature on the subject, consulting the SRs that deepened the methodological aspects of the studies examined, with attention to the characteristics of the settings implemented and the terminology used. We believe that this work can be useful in comparing the characteristics of the many AAIs described in the scientific literature and, for this reason, it is aimed at all operators involved in AAI research. Moreover, establishing consistency among the terminology and methodological approaches of these interventions could provide further useful support to clinical studies and researchers, as well as starting a new discussion in field of AAIs. Finally, there have been no URs with this objective, except for the work of Stern and Chur-Hansen [28] which aimed to explore SRs related to equine-assisted interventions (EAIs) specifically.

## 2. Materials and Methods

This study was carried out following Aromataris and Munns [29] in the Joanna Briggs Institute (JBI) Manual, to realize an umbrella review following the “Preferred Reporting Items for Systematic Reviews and Meta-Analyses (PRISMA)” guidelines [30]. Currently, URs are rapidly spreading as a fast and effective means of spreading and presenting evidence content in medical knowledge [28]. In the literature, however, it has been clearly indicated that, in order to carry out solid scientific works, it is necessary that the operating protocols are clearly specified, the variables of interest are clearly defined, the effects of the SRs on the scientific community are indicated, the results are clearly reported, that software is used appropriately, and the limits of the work done are underlined [27]. Therefore, the study procedures were defined first in an operational protocol that specified the research strategies, the inclusion and exclusion criteria, and data extraction.

### 2.1. Inclusion and Exclusions Criteria

In this UR, although we also explored the possibility of a gray literature search, only SRs in the English language published in international peer-reviewed and high-impact indexed journals were included, in order to ensure a higher quality of results. The subject area and research domain were indicated.

Furthermore, only papers published during last six years (2013–2019) were selected, to ensure a more recent overview of the scientific literature.

In terms of content, both qualitative and quantitative SRs were included, but only those containing information about the terminology explored in SRs (e.g., AAI, AAA, or AAT), in which the terminology was considered eligible and the methodological aspects used to realize interventions were examined (e.g., frequencies and length of sessions; duration of treatment; users, animals, and operators involved).

### 2.2. Search Strategy

Our research was conducted following the three-phase search process recommended in the manual for umbrella reviews of the Joanna Briggs Institute (JBI) [29]. Papers were collected by searching on the PubMed [31], Cochrane [32], and Google Scholar [33] search engines (JBI First step).

In order to define the search query, we added (in the final strings) each of the following terminologies about the animals and the main animal species involved: Dog/Equine/Animal. We combined these with the following terms which refer to the kind of interventions and to related methodologies: Intervention/Activity/Therapy/Education/Coaching/Counseling. All terms were selected based on international reference guidelines [1]. In addition, terms relating to the involvement of Dog and Horses were included, considering that these species are the most involved in such interventions [1].

In the PubMed database, we inserted the term “Meta-Analysis [ptyp] OR Systematic [sb]”, to select only SRs. The same search strategy was adapted for the other databases examined, and is available from the authors upon request (JBI Second step).

### 2.3. Study Selection and Data Extraction

The information was extracted from each SR included in the UR in order to achieve the goal. All data were entered into an Excel data set. Data relating to terminologies used, reference disciplines, animal species and operators involved, and variations of the settings were collected. Additional data were extracted to facilitate identification of the study (i.e., first name, year of publication, journal).

The search query identified 57 articles (11 in PubMed, 3 in Cochrane Library, and 43 in Google Scholar). After evaluating all articles for titles and abstracts, papers were selected and, after removing duplicates, only papers published in journals indexed on Web of Sciences [34] and Scopus [35] were included (JBI Third Step). Finally, a total of 15 SRs met the inclusion criteria, plus one that was found through a hand search. Figure 1 represents the PRISMA (Preferred Reporting Items for Systematic Reviews and Meta-Analyses) flow-chart process [30] of study selection. Two researchers (A.S. and F.D.) examined the papers independently.

Moreover, the quality of the included reviews was evaluated using a score which was assigned according to the Health Evidence tool [36]. Each study was scored in the range from 0 to 10: Weak study quality if the score was four or less; medium quality, if the score ranged from 5–7; high quality, if it was in the range of 8–10. The score quantified the strength of the data in the studies included in each SR and was not an inclusion criterion. The inter-judge agreement was calculated (and independently identified by two judges) as a measure of reliability, assessed by Cohen’s kappa. Every disagreement was solved by intervention of the senior author (A.S.).

The 15 SRs included in the results are indicated by the name of the first author and the year and are listed in order of recency; the full references will be reported among those in the bibliography, indicated with an asterisk.

## 3. Search Results

### 3.1. Process of Selection and Inclusion of Studies

The following flowchart (Figure 1) shows the process and the criteria for inclusion of the final results.

### 3.2. Summary of Results

The results highlight how most of the SRs were published in journals belonging to the medical area and analyzed studies generally aimed at users with mental disorders. Nevertheless, in many cases, it was difficult to detect the correspondence between the terminologies explored, those used in the studies considered eligible, and the methodological aspects described (e.g., number and length of sessions, duration of treatment). This information often appeared to be interchangeable or superimposable. Furthermore, in most studies, the species involved were dogs and horses, but it was not always clear whether the operators involved were included in a specific AAIs training.

### 3.3. Description of Results

In this section, we explore these results in more detail. Table 1 shows the subject areas, indicated by the Scopus [35] and Web of Science [34] indexed engines, to which the journals that the included SRs were published in belong.

It is clear that all of the SRs were published in journals relating to scientific disciplinary areas related to the health sector and most (about 60%) of them [37,38,39,40,41,42,43,44,45,46] belonged to the medical and health sector, while the rest 40% [47,48,49,50,51] fell under other disciplines (i.e., psychology, veterinary medicine, nursing, and occupational therapy).

In Table 2, on the other hand, the users to whom the AAIs examined in the SRs were (mainly) addressed are indicated.

The included SRs highlighted that most of the studies were aimed at patients with psychiatric conditions [37,38,39,40,41,43,44,45,47,50,51], while the others were aimed to patients with deterioration or cognitive delay [42,47,49]; in all cases, these were patients who needed or were involved in rehabilitation treatments.

In Table 3, the considerations of the methodological aspects relating to the studies examined in the SRs are indicated.

It should be noted that, in many cases, although the research object of the SRs was a specific method of intervention, studies presenting other types were also considered eligible: For example, some SRs aimed at exploring the AATs had also included works relating to AAAs or, generically, AAIs [38,39,40,41,42,43,45,46,47,48,49,50,51].

The result, therefore, has an important variability in the settings described (where indicated), whose comparison appears very complex; in fact, the duration of the treatments indicated as “AATs” ranged from four consecutive days [48] to 18 months [49]; that of the “AAAs” from three weeks to two years [48]; that of the “AAIs” from 4 to 25 weeks [51]; and that of the other modalities from ten days [41] to a year [50].

The frequency indicated in the “AATs” ranged from daily [41] to once every two weeks [38]; in the “AAAs” from three sessions a week to one every two weeks [48]; in “AAIs" ranged from daily [41] to one session per week [51]; and, in the other modalities, once or twice a week [39].

Finally, the duration of the individual sessions, in the modalities indicated as “AATs”, varied from 6 min to one whole day [37]; in the “AAAs”, from 15 [41] to 120 min [37]; in the “AAIs", from 10 [37] to 180 min [40]; and, in the other modes, from the three [48] to 240 min [50]. Table 4. shows the animal species and the operators involved, as indicated by the SRs.

The professional figures involved seem to be manifold. The most suitable were Dog/Animal Handlers [37,40,41,43,47,48,49,50] and psychologists [40,41,43,49,50]. In any case, they were not always described as having specific training in this regard, nor were the criteria by which they were chosen for the management of the interventions clear. As for the animals, however, dog was the main species involved (in all SRs, except for the one prepared by Mapes and Rosen [45]), followed by horse [37,38,42,45,46,48,49,50].

### 3.4. Quality Assessment of the Studies

Regarding the quality of the included studies (Table 5), based on the above-mentioned criteria, all chosen reviews had comprehensively good quality, as none of them had a quality score of less than 7 (moderate quality). Reliability as assessed by Cohen’s kappa was .91, indicating strong agreement between the judges.

## 4. Discussions

From the analysis of the included papers, it is possible to deduce that AAIs are widely recognized in the literature, as they are widespread in the medical sector and particularly useful in the rehabilitation field [37,38,39,40,41,42,43,45,46]. Despite this, in the SRs considered, it can be highlighted that these descriptions do not always correspond to the implementation of suitably corresponding operating practices; or, at least, to the use of univocal procedures, standardized and recorded in theoretical models recognized as indispensable for health interventions [37,38,39,49]. The authors have highlighted the widespread lack of structured research designs, definitive objectives, criteria for choosing the animals involved and the operators involved, and health protocols aimed at ensuring the safety of the participants involved [37,38,39,49].

The SRs seemed to agree on two additional aspects: Firstly, the frequent involvement of dogs and horses [40,45,46], even the ways in which their presence can facilitate the trend of the activities. These preferences could be related, for dogs, to a long history of co-evolution with humans [2,52] and, for horses, to a greater predisposition by the patients involved [53]. Another aspect of concordance between the SRs is the urgent need to structure and implement AAIs characterized by specific qualitative and quantitative studies that highlight the methodological rigor and the effects of the interventions described; all, in fact, highlighted the vast heterogeneity of the analyzed works as being a limit and puts them at risk of not being useful as a resource for the scientific community.

This concern is highly acceptable. For example, we highlight the case of interventions indicated as therapies/psychotherapies, for which, as processes for the treatment of complex clinical conditions, the importance of theoretical and methodological rigor has been widely underlined [2,16]. In fact, theoretical constructs or models are rarely indicated or, if there is a diversity of approach, it is with reference to the different clinical conditions of the patients indicated. Furthermore, in the SRs, their descriptions (where present) often appeared superimposable to those of interventions indicated as another type; in terms of number of meetings, their frequency, set objectives, and training of the operators involved. For example, therapies/psychotherapies have also been considered as interventions carried out by operators whose training is not always specified or whose duration is very short, whereas it is considered essential that they are carried out by adequately trained personnel [2,12,16] and who take time to establish a therapeutic relationship that can promote the treatment of the clinical condition involved [17].

However, it must also be said that, in some cases, although the SRs strongly emphasized how this confusion is limiting for the study of AAI, the results of many were equally heterogeneous. In several SRs, in fact, a frequent incongruity between the terminologies explored by the researchers (among the most frequent: AAIs and AATs) and the papers considered eligible was evident, describing interventions in which the terms used were often juxtaposed and whose practices did not always appear to be comparable, being very varied both in terms of organization of the setting (i.e., the frequency of the sessions varies from daily to monthly, the duration of the sessions from a few minutes to many hours) and the involvement of professional figures [38,39,40,41,42,43,51].

Furthermore, this aspect is to be considered as an important limit for the exploration of these interventions, as it makes the deepening of the scientific literature and the study of the specific AAI protocols complex, creating further difficulties in the analysis of their characteristics, their replicability and, most importantly, the effects of the techniques used on the clinical conditions treated [11,12,54].

### Limits of Our Study

This study has several limitations. First, the heterogeneity of the data collected did not allow for a meta-analysis. In fact, many SRs presented only qualitative data or did not present important information that would allow us to carry out statistical comparisons between the various studies described (e.g., divide studies according to the ages of the users). However, another limitation is that we did not include meta-analyses among our results: This corresponded to an important lack of quantitative data, which did not allow us to conduct all of the analyses proposed by the JBI guidelines (e.g., estimating a common effect size or performing a stratification of evidence) [29]. On one hand, these limitations can be considered an important index of the heterogeneity that characterizes the literature on AAIs and, on the other hand, they can provide a stimulating starting point for the realization of new research.

## 5. Conclusions

The results of this UR highlight that, within the high-impact scientific literature, bibliographic research on AAIs consider them as interventions belonging to the health area, which are particularly aimed at the rehabilitation field. However, despite their wide diffusion, the effects of such interventions on the clinical conditions examined do not seem to be univocally defined, as well as their functioning in the clinical and therapeutic setting. AAIs are mainly dealt with in the health sector (i.e., AAT), concerning the treatment of mental disorders with the particular involvement of dogs and horses, and with operators at different levels of training (although specific training for AAIs is not always indicated). Nevertheless, in this field of study, there is not always a univocal use of the terminologies used to indicate the different types of interventions carried out, despite the indications recognized at an international level [1]. At the methodological level, in particular relating to the structuring of the described settings (i.e., number and length of sessions, duration of treatment), the characteristics described in the various studies appear superimposable and there was no unequivocal correspondence with the typology of intervention indicated. These results are in agreement with the literature on the subject, in which there have been complaints of how, in AAT and AAA in the healthcare facilities, it is desirable that the animal (particularly for dogs) could be handled by a trained professional in an interspecific relationship and according to interdisciplinary principles, who is able to take charge of the animal’s health and evaluate the zoonotic risk in real-time (e.g., a veterinarian) [12,55].

This information is indicative of the widespread heterogeneity present in the literature concerning the study and implementation of AAIs. Therefore, there is a need for the development, and consequent diffusion, of protocols (not only operational, but also for study and research), which provide for a univocal use of globally recognized terminologies which facilitate comparison between the numerous experiences carried out and reported in the field.

## Figures and Tables

**Figure 1 animals-10-00759-f001:**
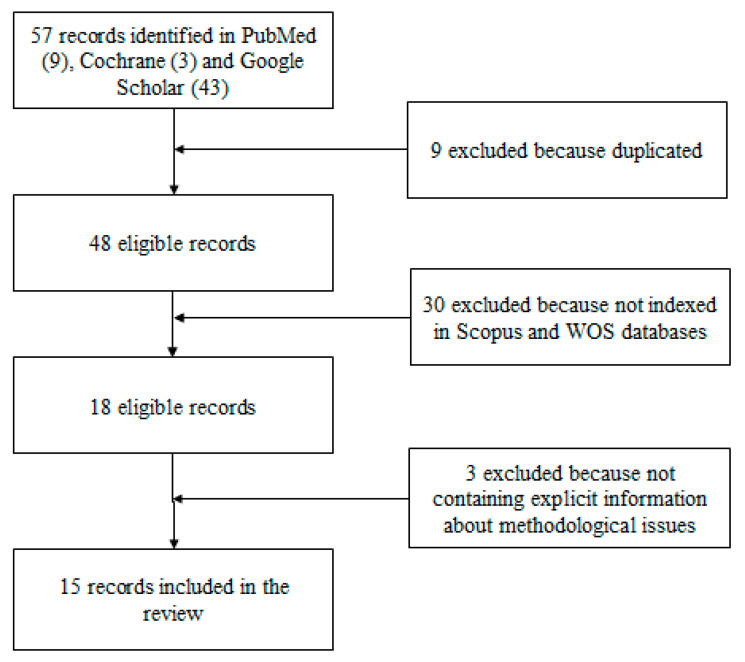
PRISMA (Preferred Reporting Items for Systematic Reviews and Meta-Analyses) process flowchart.

**Table 1 animals-10-00759-t001:** Indicated disciplinary areas in systematic reviews (SRs) included.

References	Journal	Scopus	Web of Science
		Subject Area	Research Domain
1. Hawkins, 2019	Journal of Psychiatric Research	Medicine	Psychiatry
2. Jones, 2019	PlOS One	Medicine	Science and Technology
3. Klimova, 2019	BMC Psychiatry	Medicine	Psychiatry
4. Mandrá, 2019	CoDAS	Medicine	*Not indicated.*
5. Charry-Sánchez, 2018	Complementary Therapies in Clinical Practice.	Medicine	Integrative and Complementary Medicine
6. Shen, 2018	Complementary Therapies in Medicine	Medicine	Integrative and Complementary Medicine
7. Yakimicki, 2018	Clinical Nursing Research	Nursing	Nursing
8. Brelsford, 2017	Environmental Research and Public Health	Medicine	Environmental Sciences and Ecology
9. Hoagwood, 2017	Applied Developmental Science	Psychology	Psychology
10. Bert, 2016	European Journal of Integrative Medicine	Medicine	Integrative and Complementary Medicine
11. Maber-Aleksandrowicz, 2016	Research in Developmental Disabilities	Psychology	Rehabilitation
12. Mapes and Rosen, 2016	Review Journal of Autism and Developmental Disorders	Medicine	Psychology
13. Maujean, 2015	Anthrozoos	Veterinary	Veterinary Sciences
14. O’Haire, 2015	Frontiers in Psychology	Psychology	Psychology
15. Kamioka, 2014	Complementary Therapies in Clinical Practice.	Medicine	Integrative and Complementary Medicine

**Table 2 animals-10-00759-t002:** Involved patients or users in animal-assisted interventions (AAIs), according to each SR.

References	Most Common Users	Most Common Diagnosis
1. Hawkins, 2019	Schizophrenia and related disorders *	Schizophrenia.
2. Jones, 2019	Adolescents with mental health disorders*	Physical or sexual abuse, low achievement in school, interpersonal difficulties, emotional issues, severe psychiatric illness.
3. Klimova, 2019	People with dementia *	Alzheimer’s disease.
4. Mandrá, 2019	People with Autistic Spectrum Disorder and dementia	Autistic spectrum disorder, cerebral palsy, communication disorders.
5. Charry-Sánchez, 2018	Adults with psychiatric diagnosis	Depression, dementia, multiple sclerosis, PTSD, stroke, spinal cord injury, schizophrenia
6. Shen, 2018	Children and adolescents	Various mental health disorders.
7. Yakimicki, 2018	People with dementia *	Dementia of varying stages.
8. Brelsford, 2017	Children in educational contexts *	Various emotional conditions.
9. Hoagwood, 2017	Children and adolescents with health mental problems *.	Emotional/behavioral problems, users at risk, autism spectrum disorder, attention deficit hyperactivity disorder, trauma, PTSD.
10. Bert, 2016	Children, psychiatric and elderly patients	Psychiatric diagnosis
11. Maber-Aleksandrowicz, 2016	People with intellectual disability	Mental retardation.
12. Mapes and Rosen, 2016	Children with Autistic Spectrum Disorder*.	Autistic spectrum disorder
13. Maujean, 2015	Children with Autistic Spectrum Disorder and adults with schizophrenia	Autistic spectrum disorder, schizophrenia.
14. O’Haire, 2015	Children and adolescents focused on family violence	Post-traumatic stress disorder.
15. Kamioka, 2014	People with mental and behavioural disorders	Schizophrenia, cancer, advanced heart failure, depression, ambulatory motor impairment, and older adults admitted to skilled rehabilitation units, elderly persons with chronic psychiatric, medical, and neurologic conditions.

The SRs indicated with an asterisk (*) were intended to explore only AAIs that exactly involved the type of user indicated.

**Table 3 animals-10-00759-t003:** Indicated terminologies and methodologies.

References	Terminology		Indicated Settings (If Specified)
	Explored	Considered Eligible	
1. Hawkins, 2019	AAT ^1^	AAT; AAA; PT ^6^; EAP ^7^; CAP; Novel intervention assisted by therapy dog	Treatment durations: 10–52 weeks (Therapies); 8–12 weeks (Activities; Others). Frequency of sessions: 1–7 per week (Therapies); 1–2 per week (Activities; Others). Length of sessions: 40 min to 10 hours (Therapies); 45–50 min (Activities; Others)
2. Jones, 2019	CAP ^2^	AAT; AAI; counselling	Treatment durations: 12 weeks (Therapies; Interventions). Frequency of sessions: 1 per week (Therapies; Others). Length of sessions: 45–50 min (Therapies); 180 min (Interventions; Others).
3. Klimova, 2019	Dog Therapy AAT	AAI; AAT; AAA; study with a therapy dog	Treatment durations: 2–24 weeks (Therapies); 12 weeks (Activities; Interventions). Frequency of sessions: 1–2 per week (Therapies); 2 per week (Activities; Interventions). Length of sessions: 10–45 min (Therapies); 30 min (Activities; Interventions).
4. Mandrá, 2019	AAT	PT; AAI; AAT; THR ^8^; AAA; EAA ^9^; CAT ^10^; Elephant-assisted therapy; hippotherapy; canine therapy program; dog therapy	*Not indicated.*
5. Charry-Sánchez, 2018	AAT	AAI; AAT; AAA; DAI ^11^; EAT; THR Pet-assisted living intervention; hippotherapy	Treatment durations: 3–52 weeks (Therapies); 10–12 weeks (Interventions); 3–12 weeks (Activities; Others). Frequency of sessions: 5 per week to 1 every 2 weeks (Therapies); 1–2 per week (Interventions); 2 per week (Activities); 1 per week (Others). Length of sessions: 15–180 min (Therapies); 45–180 min (Interventions); 30–45 min (Activities); 30 min (Others).
6. Shen, 2018.	AAI ^3^	AAI; PT; canine visitation therapy; pet visitation program	Treatment durations: 1–6 weeks (Therapies); 8–10 weeks (Activities); 1–12 weeks (Interventions). Length of sessions: 15–120 min (Therapies); 60 min patient-driven (Activities); 10 min patient-driven (Interventions).
7. Yakimicki, 2018	AAI	AAT; AAA; PT; DAT ^12^; pet-assisted living intervention	Treatment durations: 4 days to 1 year (Therapies); 4–12 weeks (Interventions); 3 weeks to 2 years (Activities); 6 weeks (Others). Frequency of sessions: 1–2 per week (Therapies; Interventions); 2 per week to 1 every two weeks (Activities); 3 per week (Others). Length of sessions: 10–240 min (Therapies); 15–90 min (Interventions); 30–100 min (Activities); 3–15 min (Others).
8. Brelsford, 2017	AAI	AAA; dog visitation program; therapy dog; human animal team approach; animal-assisted reading program; rabbit-assisted intervention; animal-assisted literacy instruction; human–animal intervention team model	Treatment durations: 9 weeks to 1 year school (Therapies); 10–24 weeks (Interventions); 8 weeks (Activities); 4 weeks to 1 year school (Others). Frequency of sessions: 2 per week (Therapies); 1–3 per week (Interventions); everyday (Activities); 1 per week (Others). Length of sessions: 10–90 (Therapies); 60–120 min (Interventions); 20–360 min (Others).
9. Hoagwood, 2017	AAT	AAI; AAT; AAA; therapeutic horseback riding; pet visitation; equine facilitated learning prevention program	Treatment durations: 5–20 weeks (Therapies); 12 weeks (Interventions); 12–24 weeks (Activities); 8–12 weeks (Others). Frequencies of sessions: 1–2 per week (Therapies); 1 per week (Activities; Others) Length of sessions: 30–45 min (Therapies); 10–180 min (Interventions); 60 min (Activities).
10. Bert, 2016	AAA ^4^ AAT	AAI; AAT; AAA; PT; DAI; EAP; CAP; canine-assisted ambulation; pet visitation; therapy dog	Treatment durations: 6 weeks to 3 months (Therapies); 10 weeks to 2 months (Activities). Frequency of sessions: 1–3 per week (Therapies); 1 per week to 1 per month (Activities). Length of sessions: 6 min to 1 day (Therapies); 30–120 min (Activities); 10–60 min (Interventions); 12–20 min (Others)
11. Maber-Aleksandrowicz, 2016	AAT	AAT; EAT; THR; equestrian therapy; onotherapy; kynotherapy; therapeutic animal; pet-facilitated therapy	Treatment durations: 6 weeks to 18 months (Therapies). Frequency of sessions: 1–5 per week (Therapies). Length of sessions: 7–240 min (Therapies).
12. Mapes and Rosen, 2016	EAT ^5^	THR; EAT; EAA; hippotherapy	Number of sessions: 10–70 (Therapies) (*not further specified*).
13. Maujean, 2015.	AAT	AAT; THR; AAI	Treatment durations: 10–12 weeks (Therapies); 4–25 weeks (Interventions). Frequency of sessions: 1–2 per week (Therapies); 1–3 per week (Interventions). Length of sessions: 30–180 min (Therapies); 15–50 min (Interventions).
14. O’Haire, 2015.	AAI	AAT; CAT; DAT; equine facilitated (psycho)therapy; natural horsemanship; psychiatric service dog	Treatment durations: 4–12 weeks (Therapies); 1 week to 1 year (Others). Length of sessions: 15–120 min (Therapies); 20–240 min (Others).
15. Kamioka, 2014	AAT	AAT; AAA; AAI; PT; DAT; animal facilitated therapy; service dogs; avian companionship	Treatment durations: 2–12 weeks (Therapies); 4–8 weeks (Activities); 5 days to 12 weeks (Interventions); 10 days to 4 weeks (Others). Frequency of sessions: daily to twice per week (Therapies); 1–3 per week (Activities); daily to twice per week (Interventions). Length of sessions: 12–180 min (Therapies); 15–50 min (Activities); 90–180 min (Interventions).

^1^ AAT = Animal-Assisted Therapy; ^2^ CAP = Canine-Assisted Psychotherapy; ^3^ AAI = Animal-Assisted Intervention; ^4^ AAA = Animal-Assisted Activity; ^5^ EAT = Equine-Assisted Therapy; ^6^ PT = Pet Therapy; ^7^ EAP = Equine-Assisted Psychotherapy; ^8^ THR = Therapeutic Horseback Riding; ^9^ EAA = Equine-Assisted Activity; ^10^ CAT = Canine-Assisted Therapy; ^11^ DAI = Dog-Assisted Intervention; ^12^ DAT = Dog-Assisted Therapy.

**Table 4 animals-10-00759-t004:** Involved interventionists/operators and animals.

References	Interventionists/Operators	Involved Animals
1. Hawkins, 2019	*Not specified*	Dogs, horses, farm animals, and hamsters
2. Jones, 2019	Facilitators (students, counsellors, psychologists, animal handlers), in some cases with specific training	Dogs *
3. Klimova, 2019	*Not specified*	Dogs *
4. Mandrá, 2019.	Physicians, psychologists, physiotherapists, occupational therapists, pedagogists, nurses, speech therapists, educators	Dogs, horses, fishes, guinea pigs, elephants, and insects
5. Charry-Sánchez, 2018	Therapists (not further specified)	Dogs, horses, farm animals, and cats
6. Shen, 2018.	*Not specified*	Dogs and horses
7. Yakimicki, 2018	Deliverers (animal trainers, certified therapy dog trainers, geriatric nurse practitioner, veterinarians, extended-care facilities staff and therapy dog volunteers, research staff and volunteers, dog therapy guides, staff nurses, centre staff, certified dog handlers, recreational therapy staff, recreation therapy staff and animal therapists, veterinarians, and psychiatric nurses)	Dogs, fish, and cats
8. Brelsford, 2017	Dog handlers	Dogs and guinea pigs
9. Hoagwood, 2017	Trained animal handlers	Dogs, horses, cats, rabbits, other farm animals, and guinea pigs
10. Bert, 2016	Animal handlers (not further specified).	Dogs, cats, fishes, rabbits, reptiles, and other rodents
11. Maber-Aleksandrowicz, 2016	Psychologists, equine instructors, dog-therapists, teachers, occupational therapists, therapy dog handlers, therapists, or volunteers	Dogs, horses, donkeys, and guinea pigs
12. Mapes and Rosen, 2016	*Not specified*	Horses *
13. Maujean, 2015.	*Not specified*	Dogs, horses, and farm animals
14. O’Haire, 2015.	Social workers, riding instructors, dog handlers, psychologists, veterinarians, volunteers, therapists, or researchers	Dogs, horses, cats, and farm animals
15. Kamioka, 2014	Animal handlers, psychologists, or not further specified	Dogs, cats, dolphins, birds, cows, rabbits, ferrets, and guinea pigs

The SRs indicated with an asterisk (*) were specifically aimed at exploring only AAIs involving the indicated species.

**Table 5 animals-10-00759-t005:** General description and evaluation of included reviews’ characteristics.

References	Conclusions	Limits	H.E. Score
1. Hawkins, 2019	Based on the included studies, it is not possible to confirm whether AAT ^1^ is or is not effective in treating schizophrenia as rigorous, large-scale randomized controlled trials with long-term follow-up are needed.	Included studies were heterogeneous, of lower quality, and only in the English language. Moreover, the included studies were limited to equine-assisted interventions, peer-reviewed papers, and included participants very wide age range (18–65 years).	10
2. Jones, 2019	CAP ^2^ may improve the efficacy of mental health treatments in self-selected adolescent populations by reductions in primary symptomatology (i.e., PTSD ^3^, internalizing symptoms, and the severity of serious psychiatric disorders). This non-pharmacological therapy (CAP) may also confer further benefits through secondary factors that improve therapeutic processes and quality of life (e.g., socialization). A clear nomenclature to describe the interactions between dogs, facilitators, and participants were proposed.	This work presented a heterogeneous and small number of studies. Only four of the studies achieved “fair” or “good” methodological quality plus a moderate to high level of evidence.	10
3. Klimova, 2019	The findings showed that AAT may represent a beneficial and effective complementary treatment (particularly in the area of psychological and behavioral symptoms) for patients with different levels of dementia severity.	The included studies showed different methodological approaches to AAT or AAA, with small subject samples as well as different intervention periods. Only one study measured the effect after the follow-up period.	9
4. Mandrá, 2019	A great diversity in the AAT practice was showed; in fact, performed by different professionals in the areas of health and education (mostly in the medical field), but few programs applied an interdisciplinary approach. Several animal species were involved as mediators of the therapeutic intervention, mainly dogs and horses, specifically for ASD ^4^.	Although the used programs showed positive effects in different genders and age groups of patients/users, the included studies were very heterogenous and carried out in several settings, and were different regarding the number of participants, gender, age groups, and diagnosis.	10
5. Charry-Sánchez, 2018	Despite the lack of research published in scientific journals regarding AAT for PTSD, the results suggest a potential benefit in this field. In particular, there is strong evidence supporting the use of EAT ^5^ for motor outcomes and quality of life in patients with MS ^6^, as well as in patients with stroke and spinal cord lesions.	In this review, only articles in European languages were considered. Overall quality of the articles was low. They showed a high variability regarding methodological instruments and a lack of detailed information regarding specific techniques.	10
6. Shen, 2018	The findings of current study suggest that “bodily contact” is one of the most important features contributing to the effectiveness of AAI ^7^, even across a variety of settings, as people may subjectively choose some variables compared with others (i.e., physical interaction vs. appearance).	Only seven articles in English language were included, although all had minor methodological limitations and all review findings had good quality.	9
7. Yakimicki, 2018	The majority of included studies in this review, have shown that animal-assisted interventions (AAIs) are effective in reducing the behavioral and psychological symptoms of dementia (BPSD). This study has identified several areas for continued research and refinement of these interventions. Concluding that AAIs can represent a non-pharmacological therapy for the reduction of BPSD.	The included articles showed a wide array of measured symptoms, representing a limiting factor in this systematic review. Comparison between studies was difficult. as the study designs and statistical methods used varied widely. Moreover, all studies involved a small number of participants and there was a relatively small number of RCTs ^8^.	10
8. Brelsford, 2017	The majority of the included studies reported beneficial effects on cognitive and socio-emotional behavior and physiological responses in the school setting.	Large variation in design of the included studies and several identified external factors that may have influenced the results. Sample sizes are often small, containing mixed ages or mixed abilities. Many studies did not include an adequate control group in the experimental design.	10
9. Hoagwood, 2017	AAT for children with (or at risk of) developing mental disorders represents a complementary and integrative therapeutic approach with limited but growing scientific support. Few studies suggest that, for types of problem areas such as autism and trauma, a structured therapeutic intervention with horses or dogs may determine improved functioning.	None of the included studies addressed the mechanisms of the therapeutic process. Few studies reviewed integrated theories with specific program elements or with precise measurements of outcomes. In addition, few studies specifically included a manual and, in their absence, these interventions cannot be replicated.	10
10. Bert, 2016	AAT or AAA ^9^ for hospitalized patients seem be useful and safe for a wide range of diseases, although many aspects regarding the type of intervention, safety, economic issues, and patients that would greatly benefit these programs remained unclear.	Heterogeneity and low quality of the retrieved studies, and only few works were RCTs. Most of the included papers presented limited samples. Some studies lacked a control group, while others were pilot studies. Only few studies described the sanitary protocols adopted for the animals involved in detail. Finally, some papers lacked details of randomization or considered only parent or patient opinions.	9
11. Maber-Aleksandrowicz, 2016	The evidence provided in this paper highlights that AAT may be a potentially useful supportive intervention in improving quality of life in persons with intellectual disability, although good quality research is yet lacking.	This review included AAT studies having a targeted population exclusively to ID ^10^, to excluding studies with mixed populations. Moreover, only peer-reviewed published journal articles were included (i.e., excluding gray literature).	10
12. Mapes and Rosen, 2016	This review paper represents a starting point for future research, in order to determine the validity and reliability of EAT for children with ASD. In fact, this work could be useful for researchers in order to identify the most effective research designs and settings in this field.	In the research studies on EAT, small sample sizes were included due to cost, the associated challenges of data collection in real-life situations, and the use of live animals in research (i.e., concordance with ethical guidelines for animal use). In addition, the included studies showed a lack of randomization or control groups, low capability of replication, and low standardization.	10
13. Maujean, 2015	AAIs may be of benefit to a wide range of individuals, including children with ASD and adults with psychological disorders; particularly in schizophrenia.	The included studies were performed using relatively small sample sizes (21–99 patients). In fact, all except one showed a statistical power analysis that confirmed a sample size not useful for detecting an effect.	9
14. O’Haire, 2015	As reported by Authors, AAI shows promise as a complementary technique, but it is necessary to carry out further research to better understand the different aspects of its beneficial effects in primary treatment for trauma.	Assessments in the included review studies were predominantly self-reported. Published and unpublished work were included. In both categories, positive outcomes were reported; although, the effects in published studies were greater than those in unpublished studies. Another potential bias was researcher “expectancy bias” as, in some studies, the researcher designed and performed the study in addition to presenting the intervention.	10
15. Kamioka, 2014	AAT may represent an effective treatment for several illness conditions, such as mental and behavioral disorders, such as depression, schizophrenia, and alcohol/drug addictions, based on a holistic approach of interaction with animals in nature.	Only studies with English and Japanese key words were searched and included. A relatively small and heterogeneous sample of studies was included. The standard procedures for estimating the effects of moderating variables were not followed.	10

^1^ AAT = Animal-Assisted Therapy; ^2^ CAP = Canine-Assisted Psychotherapy; ^3^ PTSD = Post Traumatic Stress Disorder; ^4^ ASD = Autism Spectrum Disorder(s); ^5^ EAT = Equine-Assisted Therapy; ^6^ MS = Multiple Sclerosis; ^7^ AAI = Animal-Assisted Intervention; ^8^ RCTs = Randomized Controlled Trials; ^9^ AAA = Animal-Assisted Activity; ^10^ ID = Intellectual Disability.

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
