# Peer review of "Methodological and Terminological Issues in Animal-Assisted Interventions: An Umbrella Review of Systematic Reviews"

_animals, 2020, doi:10.3390/ani10050759_

Round 1

Reviewer 1 Report

A very interesting review and an outstanding attempt to highlight the importance of AAI in promoting wellbeing and health. This field of research and intervention needs more and more attention in terms of standardized procedures, and this systematic review represents an advancement on the topic. I suggest it for publication without revisions.

Author Response

Dear Reviewer,

We thank you very much for the time you have taken to revise our paper and to have approved it without revisions.

Reviewer 2 Report

I consider it interesting from a scientific point of view to carry out a review. However, I believe that the study needs to be significantly improved in order to be published.

I attach the document with my comments.

Reviewer 3 Report

Brief summary

Thank you for the opportunity to review this interesting paper. The major strength is that the review includes current evidence from an important and stimulating research field. The major weaknesses are that a clearly stated review question is missing and that it seems as no critical appraisal of included studies were conducted (which is considered a key element in umbrella reviews). A major consideration is whether an umbrella review was the most appropriate choice of method. A key feature of umbrella reviews is that the review should result in evidence about a given topic or question, and it is not clear what evidence the authors of the current umbrella review are heading for.

Comments to the authors

Title and author information

The title is informative. However, the expression “Methodological and terminological issues” seems a bit vague. In addition, there are no clearly stated review questions in the paper. If clear review questions are formulated those could serve as a basis for refinement of the title. 

Introduction

The introduction is comprehensive and covers the main elements of the topic. It covers the extant knowledge addressing the question of the umbrella review. The reason for undertaking the umbrella review is stated. However, the target audience is not clearly stated.

Aims and review questions

The study objected to explore the scientific literature regarding methodological aspects useful for the realization of AAIs.

The review also aimed to compare the results and to determine characteristics of AAIs, in different contexts.

Thus, the review question seems to be rather broad and it is not evident what the authors include in the term “methodological aspects”. In addition, there is no mention of terminological issues as stated in the title.

To determine the consistency between title, aim, and inclusion criteria a clearly stated review question would be helpful.

Inclusion criteria

The inclusion criteria were:

Systematic reviews in English published in international peer-reviewed journals with references to methodological aspects of Animal Assisted Interventions, the nomenclatures used, and the animal species and operators involved.

However, in accordance to the manual from Joanna Briggs Institute should be more detailed. Foremost, it should be directed by a clearly stated review question. So, it would be helpful with a clearly stated review question and clearly stated inclusion criteria that are consistent with the study’s aim and review question(s). Could you please add more details? For example, definitions of different groups of participants, or details of settings/contexts, and clear definitions of interventions. Did the umbrella review include both qualitative and quantitative systematic reviews?

What were the pre-defined exclusion criteria?

Search strategy

The search strategy should address each component of the review question. However, it is difficult to assess the search strategy since there is no clearly defined review question.

Why was the search period limited to the last six years?

Did the search strategy involve any predefined search filters for reviews?

Apparently, the search strategy involved the keyword “review”. Did you try to use other keywords such as “systematic” or “meta- analysis”?

Was the search conducted in line with the three-phase search process recommended in the manual for umbrella reviews from the Joanna Briggs Institute? If, so could you please describe the content in the three steps? If not, could you please expand a bit on the reasons for that?

Did the authors conduct a search for gray literature?

Study selection

Was the study selection based on title and abstract examination or on full text examination? Was there a review protocol containing pre-specified inclusion criteria? How many reviewers conducted the study selection? How were disagreements between reviewers solved?

Assessment of methodological quality

How was methodological quality assessed?

Was a critical appraisal of the identifies reviews conducted? If so, could you please provide detailed information of the items that were used for the appraisal? How many members of the review team conducted the critical appraisal? How were decisions made regarding study quality and eligibility?

Data collection/extraction

Under 2.2, the second paragraph about data extractions, seems to fit better under 2.3.

Was any data extraction tool used? What measures were taken to avoid risk of bias? If not, what criteria guided the data extraction?

The flow chart (Figure 1) should be placed in the Results section together with a narrative summary of the search results and selection process and results. 

Data summary/results

I find the Result section a bit difficult to follow. I would like to recommend that it is reversed in accordance with the Joanna Briggs Institute manual chapter 10.3.8.

According to Joanna Briggs Institute a key feature of an umbrella review is a synthesis of the evidence. The synthesis should be appropriated for the review question. Since a clearly stated review question is missing this is difficult to assess. However, I cannot find a clearly stated synthesis that aims to clarify “methodological and terminological issues in in Animal Assisted Interventions”. It would be very helpful with a short summary of the main findings relation to the study objective and review question(s).

Conclusions

The authors conclude there is a need for development, and consequent diffusion, of protocols for practice, study and research. In addition, the authors state it is desirable that the animals involved in AAI are handled by trained professionals and veterinarians that can take into charge the animal’s health and can evaluate zoonotic risks.

This is not new to us who are in the field. The conclusions may need to be revised if a clear review question is formulated.

Round 2

Reviewer 2 Report

The changes made are satisfactory.

Author Response

We would like to thank the Reviewer for approving the corrections made to our manuscript.

Kind Regards,

Antonio Santaniello

Reviewer 3 Report

I thank the authors for being very attentive to my previous comments. The manuscript has improved substantially. The manuscript now reads well and is much easier to follow. Well done.

Author Response

We would like to thank the Reviewer for approving the corrections made to our manuscript and for its very positive evaluation.

We will also improve the English language and style, as you suggest.

Kind Regards,

Antonio Santaniello